# Intermittent BRAF inhibition in advanced BRAF mutated melanoma results of a phase II randomized trial

Maria Gonzalez-Cao [1✉], Clara Mayo de las Casas[1], Juana Oramas[2], Miguel A. Berciano-Guerrero [3], Luis de la Cruz[4], Pablo Cerezuela[5], Ana Arance[6], Eva Muñoz-Couselo[7], Enrique Espinosa[8], Teresa Puertolas[9], Roberto Diaz Beveridge[10], Sebastian Ochenduszko[11], Maria-Jose Villanueva[12], Laura Basterretxea[13], Lorena Bellido[14], Delvys Rodriguez[15], Begoña Campos [16], Clara Montagut[17], Ana Drozdowskyj[1], Miguel A. Molina [1], Jose Antonio Lopez-Martin [18,20] & Alfonso Berrocal[19,20✉]

Combination treatment with BRAF (BRAFi) plus MEK inhibitors (MEKi) has demonstrated survival benefit in patients with advanced melanoma harboring activating *BRAF* mutations. Previous preclinical studies suggested that an intermittent dosing of these drugs could delay the emergence of resistance. Contrary to expectations, the first published phase 2 randomized study comparing continuous versus intermittent schedule of dabrafenib (BRAFi) plus trametinib (MEKi) demonstrated a detrimental effect of the "on—off" schedule. Here we report confirmatory data from the Phase II randomized open-label clinical trial comparing the antitumoral activity of the standard schedule versus an intermittent combination of vemurafenib (BRAFi) plus cobimetinib (MEKi) in advanced BRAF mutant melanoma patients (NCT02583516). The trial did not meet its primary endpoint of progression free survival (PFS) improvement. Our results show that the antitumor activity of the experimental intermittent schedule of vemurafenib plus cobimetinib is not superior to the standard continuous schedule. Detection of *BRAF* mutation in cell free tumor DNA has prognostic value for survival and its dynamics has an excellent correlation with clinical response, but not with progression. NGS analysis demonstrated de novo mutations in resistant cases.

---

[1] Translational Cancer Research Unit, Instituto Oncologico Dr Rosell, Dexeus University Hospital, Barcelona, Spain. [2] Hospital Universitario de Canarias, Tenerife, Spain. [3] Hospitales Universitarios Regional y Virgen de la Victoria (HURyVV). Instituto de Investigaciones Biomédicas de Málaga (IBIMA), Málaga, Spain. [4] Hospital Universitario Virgen Macarena, Sevilla, Spain. [5] Hospital Clínico Universitario Virgen de la Arrixaca, Murcia, Spain. [6] Hospital Clinic, Barcelona, Spain. [7] Hospital Valle Hebron, Barcelona, Spain. [8] Hospital Universitario la Paz, CIBERONC, Madrid, Spain. [9] Hospital Miguel Servet, Zaragoza, Spain. [10] Hospital Universitario la Fe, Valencia, Spain. [11] Hospital Universitario Dr Peset, Valencia, Spain. [12] Hospital de Vigo, Pontevedra, Spain. [13] Hospital Universitario de Donostia, Guipuzkoa, Spain. [14] Hospital Universitario de Salamanca, Salamanca, Spain. [15] Hospital Insular Las Palmas, Las Palmas de Gran Canaria, Spain. [16] Hospital Lucus Agusti, Lugo, Spain. [17] Hospital del Mar, IMIM, CIBERONC, Barcelona, Spain. [18] Hospital 12 de Octubre, Madrid, Spain. [19] Hospital General de Valencia, Valencia, Spain. [20] These authors contributed equally: Jose Antonio Lopez-Martin, Alfonso Berrocal. ✉email: mgonzalezcao@oncorosell.com; berrocal.alf@gmail.com

Three different combinations of B-Raf proto-oncogene, serine/threonine kinase inhibitors (BRAFi) with mitogen-activated protein kinase inhibitors (MEKi) have been approved for the treatment of *BRAFV600* mutation-positive advanced melanoma[1–3]. This therapy has demonstrated high anti-tumor activity with fast responses in most patients. However, tumor relapse commonly occurs 12−18 months after initiation of treatment due to the emergence of multiple acquired mechanisms of resistance. One of the key pre-clinical observations in resistant tumors was that they suffered a fitness deficit in the absence of the drug[4]. Results of the analysis of two different patient-derived xenograft models treated with the BRAFi vemurafenib following an experimental schedule of four weeks on, two weeks off, showed that this regimen controlled tumor growth over the course of seven months of treatment, while mice treated on a continuous schedule developed resistance after two months[4]. Similarly, Callahan and colleagues described the case of a melanoma patient in which an intermittent schedule of vemurafenib achieved a long tumor response[5]. These findings led to the hypothesis that modulation of drug pressure through an intermittent dosing could delay the emergence of resistance and clinical studies were subsequently initiated. Contrary to expectations, the first published phase 2 randomized study comparing continuous versus intermittent schedule of dabrafenib (BRAFi) plus trametinib (MEKi) demonstrated a detrimental effect of the "on−off" schedule[6]. All patients in this study received continuous combined treatment for a 8-weeks lead-in period, after which patients with non-progressing tumors were randomized to either continuous or intermittent dosing of both drugs on a 3-week-off, 5-week-on schedule. Continuous dosing yielded a statistically significant improvement in post-randomization progression-free survival (PFS) compared with intermittent dosing (median PFS 9.0 vs. 5.5 months, $P = 0.064$)[6]. Biologic explanation of these results is unclear.

Here, we present the results of a similar phase 2 randomized study, coordinated by the Spanish Melanoma Group. The primary objective was to evaluate the anti-tumor efficacy, in terms of PFS, of continuous vs. intermittent administration of a BRAFi, vemurafenib, in combination with a MEKi, cobimetinib. An exploratory translational sub-study in cell-free DNA (cfDNA) from serial plasma samples was pre-planned as a secondary endpoint.

## Results

**Clinical activity and toxicity.** The study included 70 treatment naïve patients with advanced melanoma (Fig. 1 and Supplementary Table 1), which were randomized 1:1 to a standard arm A versus an experimental arm B. In arm A, patients received a continuous schedule, with daily vemurafenib during four-week cycles and daily cobimetinib for three weeks on and one week off. In arm B, the schedule consisted of three initial standard cycles followed by intermittent dosing with an off-treatment interval of two weeks for vemurafenib and three weeks for cobimetinib (Supplementary Methods and Supplementary Fig. 1). Median PFS in arms A and B was 16.2 months (95%CI = 9.5−24.1) vs. 6.9 months (95%CI = 5.2−9.3) ($p = 0.079$), respectively (Fig. 1A). No statistically significant differences were observed in overall survival (OS) (Supplementary Fig. 2). In the continuous arm, 25 (71.4%) patients had an objective response, including 8 (23%) complete responses (CR); while in the intermittent arm 21 (60%) patients showed an objective response, with 5 (14%) CRs. Treatment-related adverse events were in the range of those expected, with G3-4 toxicity in 42.8% of patients in the standard arm versus 39.9% in the intermittent arm. Dose reduction requirements were similar for both arms (Supplementary Tables 2, 3).

**cfDNA analysis.** An exploratory analysis of *BRAFV600* mutation in cfDNA samples was performed in 36 patients (Supplementary Methods). Twenty-one (62%) patients had detectable *BRAFV600* mutation in pretreatment cfDNA (preBRAF+). As expected[6–9], preBRAF+ patients had a significantly worse outcome compared with patients with baseline BRAF mutation undetectable in cfDNA (preBRAF−). Median PFS was 8.2 months for preBRAF+ (95%CI = 5.2−13.6) vs. non-reached (NR) for preBRAF− (95% CI = 2.8−NR) ($p = 0.017$) (Supplementary Fig. 3A), while OS was 14.7 months (95%CI = 8.5−23.6) vs. NR (95%CI = 32.6−NR) ($p = 0.0024$) (Supplementary Fig. 3B). Significant differences were found according to treatment arm, mainly in preBRAF+ patients, while among preBRAF− some patients had a long response despite receiving treatment with the on−off schedule (Fig. 2B, Supplementary Tables 4, 5 and Supplementary Fig. 4). In the continuous arm, PFS was 13.3 months (95%CI = 4.6−NR) for preBRAF+, and NR (95%CI = 2.3−NR) for preBRAF− ($p = 0.003$); while in the intermittent arm, median PFS was 6.2 months (95% CI = 0.3−8.3) vs. NR (95%CI = 2.8−NR), for preBRAF+ and preBRAF−, respectively ($p = 0.003$) (Fig. 3). Most preBRAF+ patients had high basal LDH levels (n 14/21). Prognosis of preBRAF+ patients was significantly different between patients with normal (n 7/21) or high (n 14/21) LDH levels vs. preBRAF− patients with normal LDH (n 12/13): median PFS was 7.9 months (95% CI = 2.5−13.6), 8.2 months (95% CI = 4.3−NR) and NR (95% CI = 5.3−NR), respectively ($p = 0.011$) (Supplementary Tables 6−9 and Supplementary Figs. 5−7).

In all preBRAF+ cases, *BRAF* mutation became undetectable at tumor response (Fig. 2 and Supplementary Figs. 7, 8). Among preBRAF- patients, *BRAF* mutation continued undetectable during tumor response (Fig. 3 and Supplementary Fig. 7). Twenty-three patients progressed during follow-up, 11 from arm A and 12 in arm B. In arm A, *BRAF* mutation emerged at progression in 6/11 (54.5%) cases, with a median time to *BRAF* relapse in blood of 46 weeks (range 16−84 weeks). Regarding arm B, *BRAF* mutation became detectable at progression in the blood of 9/12 patients (75%), in most cases shortly after the start of the on−off treatment, being the median time to *BRAF* relapse of only 22.5 weeks (range 10−54 weeks) (Fig. 3 and Supplementary Figs. 7, 8). In summary, although the correlation with the clinical response was good and *BRAF* mutation disappeared from cfDNA in all cases at response, 34.8% of patients remained negative in cfDNA at progression, suggesting mechanisms of resistance arising from *non-BRAFV600* cell clones. In this regard, targeted NGS analysis of eight cases at radiological progression revealed mutations in *NRAS*, *KRAS*, *PIK3CA,* and *TP53*, as well as amplifications in *BRAF*, *PDGFRA,* and *KIT* (Supplementary Table 10).

## Discussion

Our results fail to support an advantage of an intermittent schedule of BRAFi plus MEKi. Moreover, the current findings suggest that the superiority of continuous dosing may be a class effect of BRAF/MEK inhibitors and reject the previous hypothesis from Algazi et al. posing that the long half-life of trametinib may have blunted the drug withdrawal effect[6]. In our study, as half-life elimination of vemurafenib and cobimetinib are approximately 72 and 48 h, respectively, the full elimination of both drugs is anticipated within two weeks, so the intermittent schedule is causing a subsequent intermittent drug withdrawal in plasma. These results come to confirm the detrimental effect of intermittent dosing, contrary to previous results in animal models arguing for the benefit of a drug holiday over the MAPK pathway inhibition.

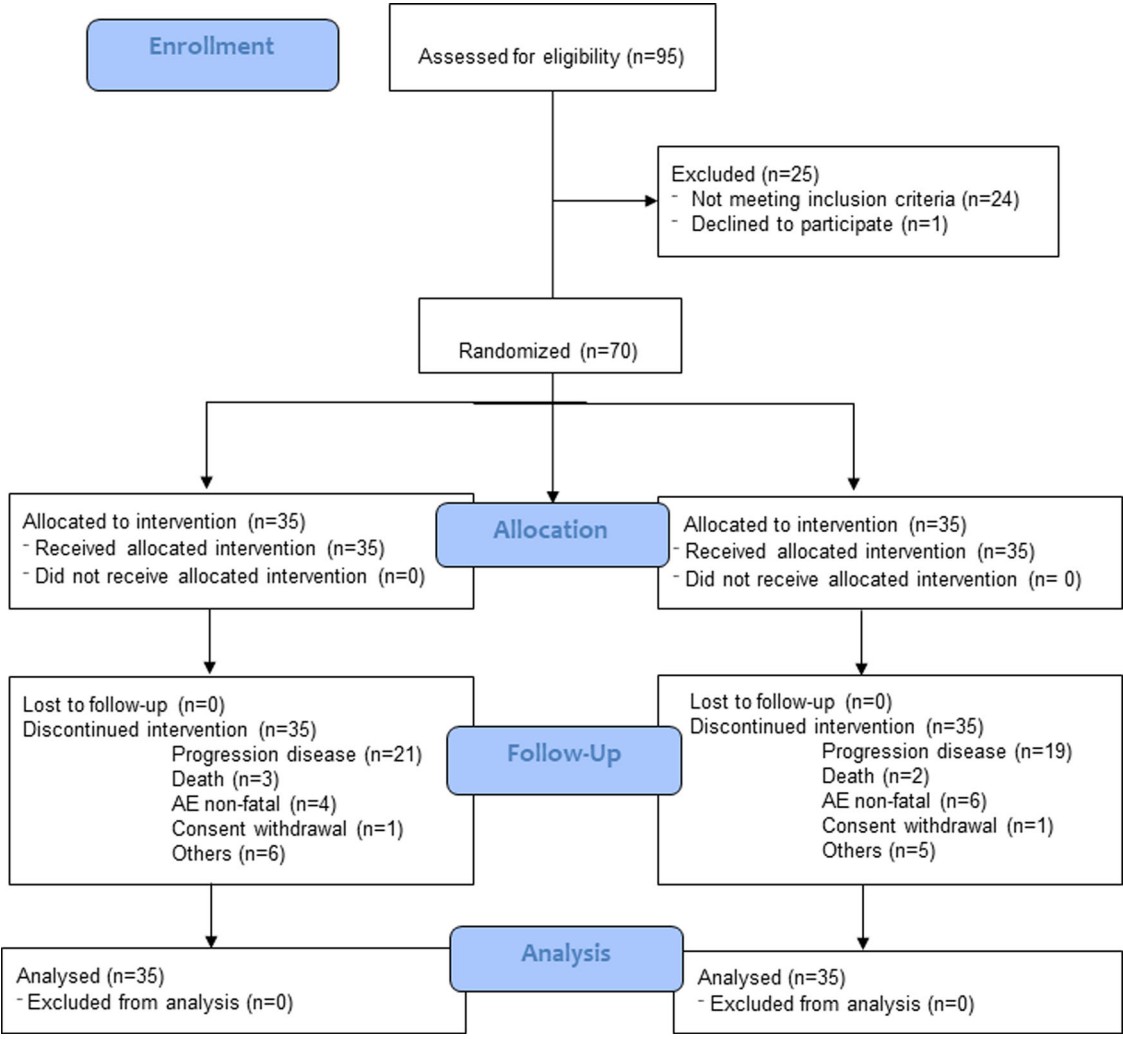

**Fig. 1** Consort flow diagram.

Finally, here we show that identification of BRAFV600 mutation in pretreatment cfDNA is associated with a dismal prognosis, with striking differences between treatment arms mainly in pre-BRAF+ patients. Clinical utility of *BRAF* testing in pretreatment cfDNA could be particularly helpful for patients with pretreatment normal LDH levels because positive results of *BRAF* testing identifies patients with a dismal prognosis into this subgroup. Interestingly, while serial monitoring of *BRAF* mutation in cfDNA throughout the treatment has a good correlation with clinical response, *BRAF* testing does not capture disease progression in a significant number of patients, mainly in those treated with the standard schedule. Understanding the role of *BRAF* mutant clones in melanoma resistance to BRAF inhibition is key to conduct rational drug development in this field. The identification of different mechanisms of resistance in plasma samples at progression could be helpful for guiding the research on novel targeted agents as salvage therapy for every individual case.

## Methods

**Study design and patients**. This is a multicenter, randomized, open-label phase 2 trial in advanced melanoma patients. The study was conducted in 19 hospitals in Spain from the Spanish Melanoma Group (GEM). Safety was assessed in all patients. Eligible patients were comprised of *BRAFV600* mutant melanoma untreated patients with stage IV or unresectable IIIc. The protocol, informed consent forms (ICF), and any appropriate related documents were submitted to the Institutional Review Board (IRB) or Independent Ethics Committee (IEC) by the principal investigator (PI) for approval. The protocol was approved by the CEIm Parc Salut Mar (C/Aiguader, 88.

Piso 1°. Edificio PRBB 08003 Barcelona). The study was initiated after the PI and GEM as the sponsor or Designee received IRB or IEC approval of the protocol and ICF. All protocol amendments were reviewed and approved by the IRB or IEC before implementation. The investigator submitted periodic reports and informed the IRB or IEC of any reportable adverse events (AEs) as per the International Conference on Harmonization (ICH) of Technical Requirements for Registration of Pharmaceuticals for Human Use guidelines and local IRB or IEC standards of practice. This study was conducted under standard operating procedures (SOPs) of the sponsor, which are designed to ensure adherence to Good Clinical Practice (GCP) guidelines as required by the following: Principles of the World Medical Association Declaration of Helsinki (2004 revision), ICH E6 Guideline for GCP (CPMP/ICH/135/95) of the European Agency for the Evaluation of Medicinal Products, Committee for Proprietary Medicinal Products, International Conference on Harmonization of Pharmaceuticals for Human Use.

Before conducting the screening procedures, the investigator obtained written informed consent from each individual participating in this study after the investigator had explained to each subject or guardian/legally authorized representative, the nature of the study, the purpose, the procedures involved, the expected duration, the potential risks and benefits involved, any potential discomfort, potential alternative procedure(s) or course(s) of treatment available to the subject, and the extent of maintaining the confidentiality of the subject's records. Each subject was informed that participation in the study was voluntary.

Eligible patients were patients with histologically confirmed melanoma, either unresectable stage IIIc or stage IV metastatic melanoma. Patients must be naïve to treatment for locally advanced unresectable or metastatic disease. Documentation of *BRAFV600* mutation-positive status in melanoma tumor tissue was required. Other inclusion criteria included measurable disease per RECIST v1.1 and ECOG performance status of 0 or 1. Additionally, patients to be included in the biomarker substudy should consent to provide archival tissue for biomarker analyses and consent to undergo tumor liquid biopsies (blood samples). The date of first patient enrollment was the 30th of June, 2015. The date of last patient enrollment was the 19th of September, 2017.

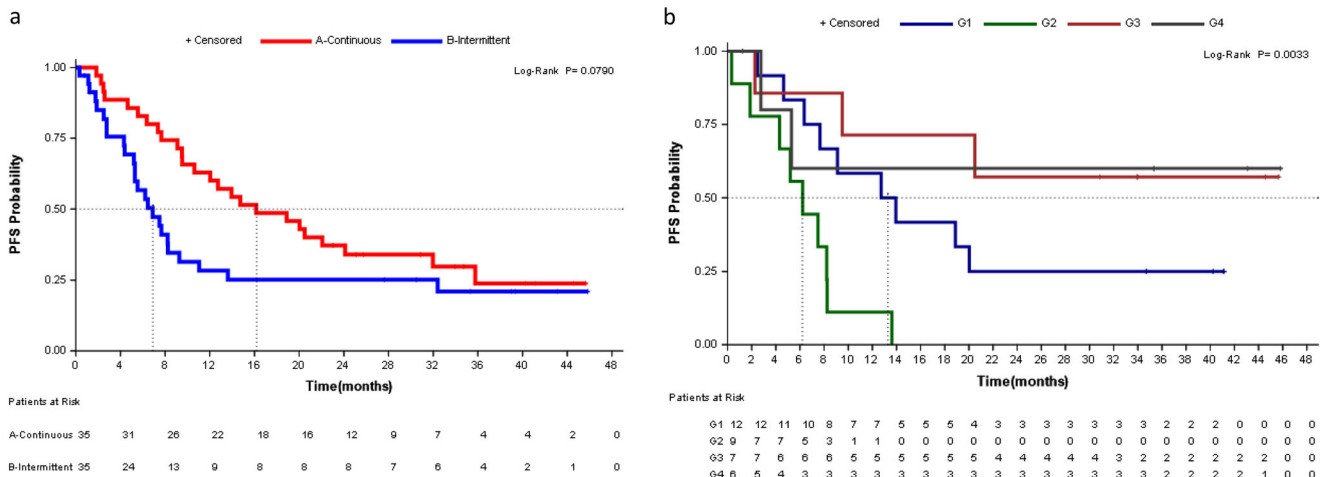

**Fig. 2 Progression Free Survival. A** PFS according to treatment arm (n 70); **B** PFS according to treatment arm and BRAF mutation in cfDNA (n 34). G1: Continuous arm and basal BRAF Positive in cfDNA, G2: Intermittent arm and basal BRAF Positive in cfDNA, G3: Continuous arm and basal BRAF Negative in cfDNA, G4: Intermittent arm and basal BRAF Negative in cfDNA.

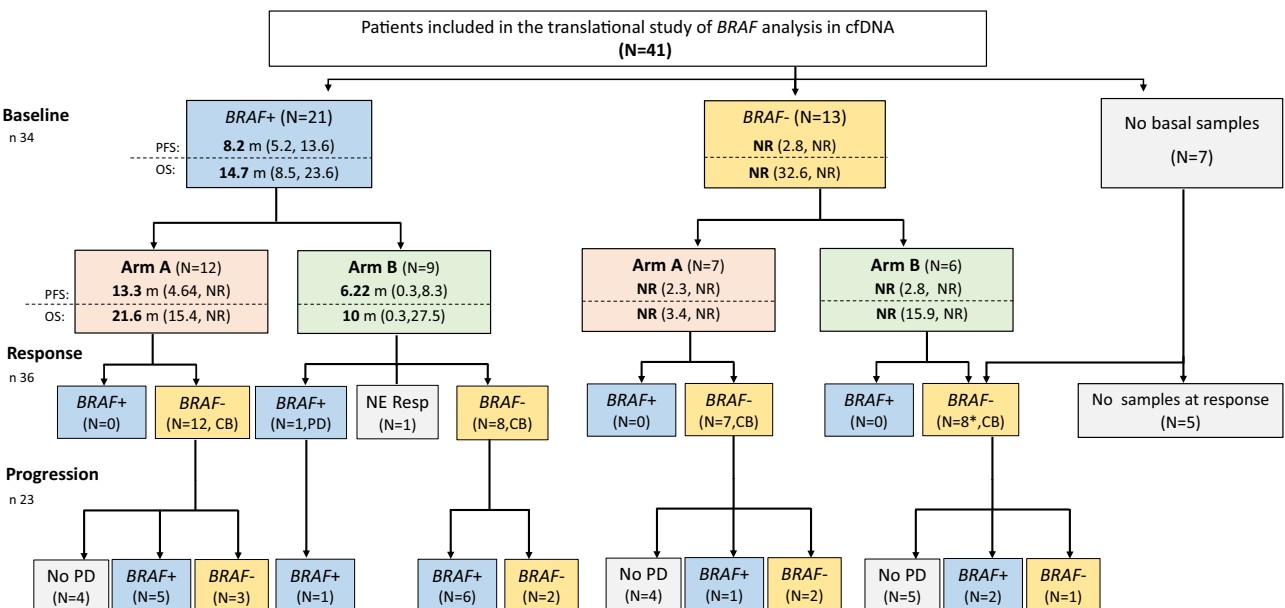

**Fig. 3 Summary of BRAF cfDNA results.** Arm A: Continuous arm; Arm B: Intermittent arm; CB: objective response or stable disease; m: months; NE resp: no response evaluation; No PD: patients without progression at data cut-off analysis; NR: no reached; PD: progression disease; *including two patients without basal samples.

**Randomization**. At the time of enrollment, patients were stratified with a 1:1 ratio according to ECOG functional status (PS 0 vs 1). LDH levels (normal vs elevated), age (65 years or older), and staging (IIIc or IV). Stratification and randomization were centralized, and the allocation was done automatically using randomized permuted blocks. The data center, randomization number, and treatment group of each patient were submitted to the investigator. The assigned treatment started within a maximum period of 1 week from randomization.

**Statistical analysis**. Sample size calculation: Using the method described by Brookmeyer R, for an error $\alpha = 0.1$ and an error $\beta = 0.20$, it will be necessary to include 34 evaluable patients per treatment group, using Log-rank (Mantel−Cox) 2-sided test. With this sample size, we would have a power of 80% to detect a difference of 23% in the percentage of patients free of progression to 1 year (with an error $\alpha = 0.1$). Progression-free survival and overall survival were estimated by means of the Kaplan–Meier method and the nonparametric log-rank test was applied for comparisons of groups. Cox semiparametric proportional-hazards model was used in the analysis of survival data to explain the effect of explanatory variables on hazard rates, obtaining Hazard Ratios (HR) and their 95% confidence intervals (CI). The descriptive statistics include mean, standard deviation, median, range for continuous variables, and the number and percentages for categorical

variables. Association analysis used the Fisher exact test or Chi-square test for categorical variables. In case of continuous variables t-test Anova or the non-parametric Wilcoxon (Mann−Whitney), as applicable. Exploratory analysis of the relation between cfDNA BRAF expression values and treatment response during the study was achieved with graphical display of results. Each analysis was performed with the use of a two-sided 5% significance level and a 95% CI. The statistical analyses were performed using SAS version 9.4.

**Procedures**. Patients were randomized to one of the following treatment regimens: Group A (continuous administration) vemurafenib 960 mg p.o. twice daily on days 1−28 and cobimetinib 60 mg p.o. once a day on days 1−21 of each 28-day treatment cycle. Group B (intermittent administration) vemurafenib 960 mg p.o. twice daily on days 1−28 and cobimetinib 60 mg p.o. once a day on days 1−21 of each 28-day treatment cycle for 12 weeks. Then, both drugs were administered at the same doses previously indicated, but with an intermittent schedule: vemurafenib days 1−28 followed by 14 days of rest (4 weeks on and 2 weeks off), and cobimetinib days 1−21 followed by 21 rest days. (3 weeks on and 3 weeks off).

**Outcomes**. The primary objective of the study was to assess the efficacy in terms of progression-free survival (PFS) of two regimens for the administration of the

vemurafenib, cobimetinib combination (continuous and intermittent) in the first-line treatment of unresectable or advanced metastatic melanoma patients with the BRAF V600 mutation. Secondary endpoints were to evaluate the safety of the regimens and activity in terms of response rate according to RECIST 1.1 criteria and overall survival. Adverse events were classified as drug-related or unrelated, according to investigator criteria, and were graded with the use of the National Cancer Institute Common Terminology Criteria for Adverse Events (NCI CTCAE), version 4.03. Safety assessments consisted of monitoring and recording all AEs, including all Common Terminology Criteria for Adverse Events (NCI-CTCAE v4.03) grades (for both increasing and decreasing severity), and SAEs; regular monitoring of hematology, blood chemistry, and urine values; periodic measurement of vital signs and ECGs; and performance of physical examinations as detailed in the schedule of assessments. All analysis were performed by intended to treat population (iTTP).

An associated translational sub-study was performed only in patients who agreed to participate and had signed the specific informed consent. The objective of this sub-study was to analyze the prognostic and predictive value of the BRAF mutation determined in cell-free DNA (cfDNA), its value for monitoring the evolution of the disease, and to explore if it can be helpful as a non-invasive technique for the determination of molecular resistance mechanisms.

**DNA isolation**. Purification of cfDNA was performed from 4 mL of plasma using a custom protocol with the QIAsymphony® DSP Virus/Pathogen Midi Kit using a QIAsymphony robot (QIAGEN, Hilden, Germany) and following the manufacturer's instructions. The final elution volume was 50 μL per sample. For liquid biopsies with less than 4 mL, an alternative custom protocol using 1.2 mL and a final elution volume of 30 μL was used.

**Analysis of BRAFV600 in cfDNA**. Analysis of BRAFV600 mutations in cfDNA was analyzed and quantified using a PNA probe-based TaqMan assay developed in house, which can detect BRAFV600E/K in samples containing as little as 0.005% mutant DNA (copy number ratio 1: 20,000).

**NGS for mutation testing**. Next-generation sequencing (NGS) of DNA isolated from plasma was performed with the GeneReader Platform (QIAGEN, Hilden, Germany). Purified DNA (16.5 μL, ~40 ng) was used as a template to generate libraries for sequencing with the GeneReadQIAact Custom DNA Panel, according to the manufacturer's instructions. The panel is designed to enrich specific target regions in 20 selected genes frequently altered in solid cancer tumors (*ALK, BRAF, CDK4, CDK6, EGFR, ERBB2, ERBB4, FGFR1, IDH1, IDH2, KIT, KRAS, MET, NRAS, PDGFRA, PIK3CA, RICTOR, ROS1, STK11, and TP53*), including *MET* exon 14 skipping mutations. Libraries were quantified using a QIAxcel® Advanced System, diluted to 100 pg/ul, and pooled in batches of 6 (liquid biopsies) or 12 (tissues). Clonal amplification was performed on 625 pg of pooled libraries by the GeneRead Clonal Amp Q Kit using the GeneReadQIAcube and an automated protocol. Following bead enrichment, pooled libraries were sequenced using the GeneRead UMI Advanced Sequencing Q kit in a GeneReader instrument. QIAGEN Clinical Insight Analyze (QCI-A) software 1.1 was employed to perform the secondary analysis of FASTQ reads, align the read data to the hg19 reference genome sequence, call sequence variants, and generate a report for visualization of the sequencing results. Variants were imported into the QIAGEN Clinical Insight Interpret (QCI-I) web interface for data interpretation and generation of the final custom report.

**Reporting summary**. Further information on research design is available in the Nature Research Reporting Summary linked to this article.

## Data availability

The protocol (including the statistical analysis plan) and the informed consent form are available in the Supplementary Information. The processed clinical main data are available at HARVARD database. All data used in this study are available in the HARVARD database under accession code https://doi.org/10.7910/DVN/TFFSGR. The sequencing data are available under restricted access in compliance with patient consent for data sharing, access can be obtained by approval from the Spanish Melanoma Group data access committee (Contact person: Maria Gonzalez-Cao, Email: secretaria@groupgem.com). Source data are provided with this paper.

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

## Acknowledgements

The authors thank Stephanie Davis for the revision of the English language and Juan Berges (Pivotal CRO). This study was supported by the Roche Group and, in part, by a Marie Skłodowska-Curie Innovative Training Networks European Grant (ELBA No 765492). The funder had no role in the design and conduct of the study, collection, management, analysis, and interpretation of the data, preparation, review, or approval of the manuscript, and decision to submit the manuscript for publication.

## Author contributions

Dr. M. Gonzalez-Cao had full access to all the data in the study and takes responsibility for the integrity of the data and the accuracy of the data analysis. *Concept and design*: Drs M. Gonzalez-Cao, A. Berrocal, JA. Lopez-Martin, C. Mayo. *Acquisition, analysis, or interpretation of data*: Drs M. Gonzalez-Cao, C. Mayo de las Casas, J. Oramas, M.A. Berciano, L. de la Cruz, P. Cerezuela, A. Arance, E. Muñoz, E. Espinosa, T. Puertolas, R. Diez, S. Ochendustko, M.J. Villanueva, L. Basterretxea, L. Bellido, D. Rodriguez, B. Campos, C. Montagut, A. Drozdowskyi, M.A. Molina, J.A. Lopez-Martin, A. Berrocal. *Drafting of the manuscript*: Drs M. Gonzalez-Cao. *Critical revision of the manuscript for important intellectual content*: Drs M. Gonzalez-Cao, C. Mayo de las Casas, P. Cerezuela, E. Muñoz, E. Espinosa, T. Puertolas, A. Drozdowskyi, M.A. Molina, A. Berrocal. *Statistical analysis*: Drs A. Drozdowskyj and M. Gonzalez-Cao. *Obtained funding*: Drs A. Berrocal, C. Mayo de las Casas. *Administrative, technical, or material support*: Drs M. Gonzalez-Cao, A. Drozdowskyj, C. Mayo de las Casas. *Supervision*: Dr M. Gonzalez-Cao, A. Berrocal, M. A. Molina, C. Mayo de las Casas. Stephanie Davis has not received any economical compensation for this work. Juan Berges works for CRO company and has carried out the administrative and technical duties for conducting the clinical trial. We confirm that we have obtained written permission to include their name in the Acknowledgment section.

## Competing interests

The authors declare no competing interests that could directly undermine, or be perceived to undermine the objectivity, integrity, and value of the publication, through a potential influence on the judgements and actions of authors regarding objective data presentation, analysis, and interpretation.

## Additional information

**Peer review information** *Nature Communications* thanks Alain Algazi, Serigne Lo, Vinicius de Lima Vazquez, and the other anonymous reviewer(s) of this work. Peer reviewer reports are available.

