## [Peer Review File · Nature Communications]

Reviewers' Comments:

Reviewer #1:

Remarks to the Author:

In their report Gonzalez-Cao report on the second trial testing intermittent versus continuous BRAF+MEK inhibition. The trial confirms the previous data, and once more shows that all animal models arguing for a drug holiday from MAPK pathway targeting were wrong. These data are very important, as the animal model data drew so much attention that (also in my experience) many patients wanted to follow drug holidays and it was very hard to convince them not to do. The manuscript is clear and well written.

Reviewer #2:

Remarks to the Author:

This manuscript describes a 70-patient study of intermittent versus continuous dosing of vemurafenib with cobimetinib in patients BRAF mutated metastatic melanoma conducted by the Spanish Melanoma Group. A study published in Nature Medicine last year examined intermittent dosing dabrafenib and trametinib in a similar population also showing that continuous dosing was superior. These initial findings were surprising to many and a number of possible explanations were discussed including the possibility that the long half-life of trametinib may have blunted the drug withdrawal effect such that any benefit of intermittent dosing was obscured. The half-life elimination of vemurafenib is approximately 2-1/2 days and the half-life elimination of cobimetinib is about 2 days so that full elimination of both drugs is anticipated within 2 weeks, an advantage over the previous study. Furthermore, the current findings suggest that the superiority of continuous dosing maybe a class effect of these medications. Based on this, the paper could be an important addition to the medical literature.

However, the manuscript requires major revisions prior to publication. The brief report format is challenging given the volume of data presented and some improvement in summarizing these data could be helpful. The supplementary data section is 70 pages long and much of these data could be summarized more effectively or culled. For example, it might be better if the cell-free DNA findings over the course of treatment could be summarized graphically rather than including a large number of figures with individual patient data.

The comments at the end of the article summarizing the findings are very brief, perhaps by necessity given the brief report format, but there is no substantive discussion of the primary clinical findings, the superiority of continuous dosing. The cell-free DNA findings are interesting, both replicating and expanding on existing published data. A further discussion of the relatively weak predictive value of serial cell-free DNA levels in predicting disease progression would be desirable, particularly in patients on the continuous dosing arm.

Although the authors' intent is generally understandable, there are some passages that would benefit from editing for flow and clarity. For example on line 133, "At this respect, targeted NGS analysis ... showed mutations in...." Labeling of some of the figures also makes it more difficult to track the main findings. For example, Figure 1B would be easier to read if the groups were labeled more descriptively (rather than G1, G2, G3, G4). The same is true with extended data figure 5. There are also some formatting problems including some boxes that are present presumably because the authors did not designate the text font as "symbol."

Reviewer #3:

Remarks to the Author:

The paper is well justified and conducted clinical and translational trial performed to answer an important hypothesis on metastatic melanoma treatment.

Although is a high value research and worthwhile for publication, some points need to be addressed:

1. There was any crossover planned? The intermittent was significantly worse than the continuous. How do you managed this question?
2. In the Extended data eTable 1. - Patient Characteristics there is no p value to analyse the homogeneous distribution of the two groups, please provide.
3. The stratification ratio 1:1 was not applied according with the M1a or M1b and the have different prognosis? Did it affect your results? Can you show some evidence?
4. For the e GeneReadQIAact Custom DNA Panel, did you determinate a OLB for each mutation? Can you explain?

Reviewer #4:

Remarks to the Author:

This study reports the results of a randomised phase II study that aimed to assess the efficacy and safety of continuous versus intermittent schedules of administration of Vemurafenib in combination with Cobimetinib, in previously untreated BRAFV600- mutation positive patients with unresectable locally advanced or metastatic melanoma. Beyond the initial trial outcomes in the study design, authors have conducted an exploratory translational sub-study in cell-free DNA using a subset of the patients. Authors should be commended for this well designed and well conducted study and the concise and well written report. However I have few comments that I hope, will help to better interpret the results. In their study protocol (p 58), it is stated "With this sample size we would have a power of 80% to detect a 23% difference in the percentage of patients free of progression at 1 year...". However, when looking at Figure 1A, we can see that the 12 month survival rates difference is much larger than 23% (about 60%-25% =35%), in addition the observed median between the two arms (16.2 versus 6.9) resulted in a much higher median difference compared to the expected median difference of 4 months (10-6 months) in the sample size calculation. Despite these findings that I found rather interesting, the main conclusion of the study is based on the single Log-rank test p-value. A known weakness of Log-rank test is it fails (no power) if the 2 hazard cross. Even though the hazard curves are not plotted I suspect the proportional hazard assumption does not hold for the PFS survival curves in Figure 1A (for the extended eFigure 3, the OS curves are crossing consequently the hazard crossed and the proportional hazard is violated). I would suggest to formally assess the proportional hazard assumption and consequently use more powerful test when the proportional hazard does not hold.

Minor comment: please include 1-year and 2-year survival rates in Figure 1a extended eFigure 3 as planned in the SAP.

Response to Reviewers specific comments:

Reviewer #1 (Remarks to the Author):

In their report Gonzalez-Cao report on the second trial testing intermittent versus continuous BRAF+MEK inhibition. The trial confirms the previous data, and once more shows that all animal models arguing for a drug holiday from MAPK pathway targeting were wrong. These data are very important, as the animal model data drew so much attention that (also in my experience) many patients wanted to follow drug holidays and it was very hard to convince them not to do. The manuscript is clear and well written.

ANSWER: No modifications are suggested. Thank you.

Reviewer #2 (Remarks to the Author):

This manuscript describes a 70-patient study of intermittent versus continuous dosing of vemurafenib with cobimetinib in patients BRAF mutated metastatic melanoma conducted by the Spanish Melanoma Group. A study published in Nature Medicine last year examined intermittent dosing dabrafenib and trametinib in a similar population also showing that continuous dosing was superior. These initial findings were surprising to many and a number of possible explanations were discussed including the possibility that the long half-life of trametinib may have blunted the drug withdrawal effect such that any benefit of intermittent dosing was obscured. The half-life elimination of vemurafenib is approximately 2-1/2 days and the half-life elimination of cobimetinib is about 2 days so that full elimination of both drugs is anticipated within 2 weeks, an advantage over the previous study. Furthermore, the current findings suggest that the superiority of continuous dosing maybe a class effect of these medications. Based on this, the paper could be an important addition to the medical literature.

However, the manuscript requires major revisions prior to publication. The brief report format is challenging given the volume of data presented and some improvement in summarizing these data could be helpful. The supplementary data section is 70 pages long and much of these data could be summarized more effectively or culled. For example, it might be better if the cell-free DNA findings over the course of treatment could be summarized graphically rather than including a large number of figures with individual patient data.

ANSWER: eFigure 9A and 9B in the Supplementary Appendix summarizes the evolution of cell-free DNA in all patients. We have deleted most of the individual graphics and have maintained only some examples. The complete individual graphics will be shared in a persistent repository.

The comments at the end of the article summarizing the findings are very brief, perhaps by necessity given the brief report format, but there is no substantive discussion of the primary clinical findings, the superiority of continuous dosing. The cell-free DNA findings are interesting, both replicating and expanding on existing published data. A further discussion of the relatively weak predictive value of serial cell-

free DNA levels in predicting disease progression would be desirable, particularly in patients on the continuous dosing arm.

ANSWER: We have incorporated a new paragraph at the end of the manuscript discussing the primary findings and the weak predictive value of BRAF identification in cfDNA at progression:

“Moreover, the current findings suggest that the superiority of continuous dosing may be a class effect of BRAF/MEK inhibitors and reject the previous hypothesis from Algazi et al posing that the long half-life of trametinib may have blunted the drug withdrawal effect.⁶ In our study, as half-life elimination of vemurafenib and cobimetinib are approximately 72 and 48 hours, respectively, the full elimination of both drugs is anticipated within two weeks, so the intermittent schedule is causing a subsequent intermittent drug withdrawal in plasma. These results come to confirm the detrimental effect of the intermittent dosing, contrary to previous results in animal models arguing for the benefit of a drug holiday over the MAPK pathway inhibition.”

“Finally, the study confirms that identification of BRAFV600 mutation in pretreatment cfDNA is associated with a dismal prognosis, with striking differences between treatment arms mainly in preBRAF+ patients. Clinical utility of BRAF testing in pretreatment cfDNA could be particularly helpful for patients with pretreatment normal LDH levels because positive results of BRAF testing identifies patients with a dismal prognosis into this subgroup. Interestingly, while serial monitoring of BRAF mutation in cfDNA throughout treatment has a good correlation with clinical response, BRAF testing did not capture disease progression in a significant number of patients, mainly in those treated with the standard schedule. Understanding the role of BRAF mutant clones in melanoma resistance to BRAF inhibition is key in order to conduct rational drug development in this field. The identification of different mechanisms of resistance in plasma samples at progression could be helpful for guiding the research on novel targeted agents as salvage therapy for every individual case.”

Although the authors' intent is generally understandable, there are some passages that would benefit from editing for flow and clarity. For example on line 133, “At this respect, targeted NGS analysis ... showed mutations in....”

ANSWER: The language has been reviewed. We have modified the sentence in line 133 to:

“In this regard, targeted NGS analysis of eight cases at radiological progression revealed mutations in NRAS, KRAS, PIK3CA and TP53, as well as amplifications in BRAF, PDGFRA and KIT”.

Labeling of some of the figures also makes it more difficult to track the main findings. For example, Figure 1B would be easier to read if the groups were labeled more descriptively (rather than G1, G2, G3, G4). The same is true with extended data figure 5. There are also some formatting problems including some boxes that are present presumably because the authors did not designate the text font as “symbol.”

ANSWER: We have included a box in each figure clarifying the label of Figure 1B and eFigure 5

Reviewer #3 (Remarks to the Author):

The paper is well justified and conducted clinical and translational trial performed to answer an important hypothesis on metastatic melanoma treatment.

Although is a high value research and worthwhile for publication, some points need to be addressed:

1. There was any crossover planned? The intermittent was significantly worse than the continuous. How do you managed this question?

ANSWER: Although there was no cross-over planned, once results were available, all participating centers were informed to discontinue treatment in arm B patients and to start standard therapy with BRAF plus MEK inhibitors, according to the approved standard schedule. This information has been included in the Supplementary Appendix in Method section:

“Although there was no cross-over planned, once results were available, all participating centers were informed to discontinue treatment in patients in order to start standard therapy with BRAF plus MEK inhibitors, according to the approved schedule.”

2. In the Extended data eTable 1. - Patient Characteristics there is no p value to analyse the homogeneous distribution of the two groups, please provide.

ANSWER: According to the reviewer recommendation, we have calculated the p value for each patient characteristic described in eTable1. We had not included in the eTable1 this p value following the CONSORT guidelines that, as far as we know, recommend that significance testing of baseline differences in randomized controlled trials should not be performed. Please, let us know if this information should be included in the final version of the supplementary appendix.

Characteristics	Arm A (n 35)	Arm B (n 35)	Global (n 70)	Test p-value
Age – y				
Median (range)	58 (49-69)	56 (29-85)	57 (29-85)	T-Test: 0.7243
Sex – n (%)				
Women	11 (31)	22 (63)	33 (47)	Chi-Square: 0.0084

ECOG PS – n (%)				
0	19 (54)	21 (60)	40 (57)	Chi-square 0.8094
1	16 (46)	14 (40)	30 (43)	
Primary melanoma – n (%)				
Cutaneous	29 (83)	28 (79)	57 (81)	Chi-Square: 0.6911
Mucosal	0	1 (3)	1 (1)	
Acral	1 (3)	2 (6)	3 (4)	
Unknown primary	5 (14)	4 (12)	9 (13)	
Stage – n (%)				
IIIC	0	1 (3)	1 (1)	Fisher:0.5890
M1a	8 (23)	6 (17)	14 (20)	
M1b	11 (31)	8 (23)	18 (26)	
M1c	16 (46)	20 (57)	36 (51)	
LDH – n (%)				
Normal	19 (54)	20 (57)	39 (56)	Chi- Square:0.8098
Elevated (≤2x ULN)	10 (29)	9 (26)	19 (27)	
Elevated (>2x ULN)	6 (17)	6 (17)	12 (17)	
Number of metastatic sites- n (%)				
0	0	1 (3)	1 (1)	Chi-Square: 0.2690
1	12 (34.3)	6 (17.1)	18 (25.7)	
2	10 (29)	10 (29)	10 (29)	
>2	13 (37)	18 (51)	31 (44)	
Prior adjuvant therapy- n (%)				
Interferon	9 (26)	18 (51)	31 (44)	Chi-Square: 0.1511
Nivolumab	9 (26)	7 (20)	16 (22)	
Ipilimumab	1 (3)	0	1 (1)	

3. The stratification ratio 1:1 was not applied according with the M1a or M1b and the have different prognosis? Did it affect your results? Can you show some evidence?

ANSWER: We have performed an analysis of progression free survival according to tumor stage (see attached K-M graphic in the next page). Patients with stage M1a or IIIC had a median progression free survival of 12 months (95%CI 2,8 to 35,8), while patients with stage M1b had a median survival of 7.7 months (95%CI 4,3 to NR). The total number of patients with stage M1a/IIIC in each arm was similar: in arm A (continuous schedule) there were 8/35 patients with stage M1a, while in arm B (intermittent schedule) there were 7/35 patients with stage M1a/IIIC (including one case

with stage IIIc). The total number of patients with stage M1b was 10/35 in arm A and 8/35 in arm B. Survival of patients with stage M1b was numerically shorter than survival of patients with stage M1c (median progression free survival was 9,5 months (95% CI 6,2 to 13.9 for M1c).

Strata	Subjects	Event	% Events	Censored	% Censored	Median	CI 95% LL	CI 95% UL
M1a+IIIc	15	11	73.3	4	26.7	12.0	2.8	35.8
M1b	19	13	68.4	6	31.6	7.7	4.3	.
M1c	36	26	72.2	10	27.8	9.5	6.2	13.9

Test	Chi-Square	DF	Pr > Chi-Square
Log-Rank	0.3833	2	0.8256

Comparision	Chi-Square	Raw	Sidak
M1b vs M1a+IIIc	0.00681	0.9342	0.9957
M1c vs M1a+IIIc	0.2841	0.5940	0.8352

Contrast	HR pvalue	HR Estimate	CI 95% LL	CI 95% UL
M1a+IIIc vs M1c	0.6472	0.848	0.418	1.720
M1b vs M1c	0.5898	0.832	0.426	1.624

When we analyzed progression free survival by tumor stage in each treatment arm, patients in arm B (intermittent schedule) had shorter median PFS for all tumor stage subgroups than patients treated in arm A. Patients treated in arm A had a median PFS of 18m, 24 m and 18 m, for stages M1a, M1b and M1c, respectively; In arm B median PFS was 6.9m, 5.9 m and 7.5 m, for stages IIIc+M1a, M1b and M1c, respectively. (Graph 1 and 2).

Graph 1. Kaplan Meier model PFS by Tumor Stage- Treatment A-Continuous

Kaplan Meier survival model summary results arm A

Strata	Subjects	Event	% Events	Censored	% Censored	Median	CI 95% LL	CI 95% UL
M1a+IIIc	8	6	75.0	2	25.0	18.1	1.8	.
M1b	11	6	54.5	5	45.5	24.1	2.5	.
M1c	16	13	81.3	3	18.8	13.3	9.1	22.1

Log Rank

Test	Chi-Square	DF	Pr > Chi-Square
Log-Rank	1.3339	2	0.5133

Log-Rank Multiple comparisons

Comparison	Chi-Square	Raw	Sidak
M1a+IIIc vs M1c	0.4533	0.5008	0.7508
M1b vs M1c	1.3339	0.2481	0.4347

PH Cox Regression results

Contrast	HR pvalue	HR Estimate	CI 95% LL	CI 95% UL
M1a+IIIc vs M1c	0.6230	0.783	0.295	2.076
M1b vs M1c	0.2564	0.569	0.215	1.507

Graph 2. Kaplan Meier model PFS –by Tumor Stage- Treatment B-Intermittent

Kaplan Meier survival model summary results

Strata	Subjects	Event	% Events	Censored	% Censored	Median	CI 95% LL	CI 95% UL
M1a+IIc	7	5	71.4	2	28.6	6.9	2.8	.
M1b	8	7	87.5	1	12.5	5.9	1.2	32.4
M1c	20	13	65.0	7	35.0	7.5	5.2	13.6

Test	Chi-Square	DF	Pr > Chi-Square
Log-Rank	0.5008	2	0.7785

Log-rank Multiple comparisons

Comparision	Chi-Square	Raw	Sidak
M1a+IIc vs M1c	0.000919	0.9758	0.9994
M1b vs M1c	0.3026	0.5823	0.8255

PHCoX Regression results

Contrast	HR pvalue	HR Estimate	CI 95% LL	CI 95% UL
M1a+IIc vs M1c	0.9030	0.938	0.333	2.641
M1b vs M1c	0.5366	1.338	0.531	3.370

In our opinion, it is unlikely that the results of the primary end point of the study have been affected by a heterogeneous distribution of patients between the treatment arms.

We have not included this new information in the revised manuscript in order to follow Reviewer #2's recommendation related to the extension of the supplementary appendix "The supplementary data section is 70 pages long and much of these data could be summarized more effectively or culled". Please, let us know if these data should be included in the manuscript.

4. For the e GeneReadQIAact Custom DNA Panel, did you determinate a OLB for each mutation? Can you explain?

ANSWER: QIAGEN has incorporated the UMIs technology and Single primer extension (SPE)-based primer to the design of the QIAact panels, enabling digital sequencing to correct for PCR duplicates and errors and to increase sequence uniformity. In the case of the GeneRead QIAact Custom DNA panel, the LOB was estimated for each hotspot mutation in clinically relevant genes (QCI™ Analyze for GeneReader Analysis Workflow Release Documentation). To verify the specificity of the test, we used a DNA obtained from a pool of pan-negative samples derived from ten normal donors, and a total of 20 cancer cell lines previously genotyped, including three pan-negative for the hotspot mutations included in the panel. These samples were tested in different sequencing runs, using at least two batches of reagents manufactured separately. QIAGEN Clinical Insight Analyze (QCI-A) software was used to perform the secondary analysis of FASTQ reads, align the read data to the hg19 reference genome sequence, and call the sequence variants. All variants observed below 0.5% VAF were considered as artifacts. The software did not detect false positive variants in any case within the regions of interest of the hotspot mutations included in the panel. Additionally, manual inspection was carried out for the hotspot positions in genes such as, BRAF, KRAS, NRAS, PIK3CA or EGFR, among others, confirming the negative results obtained. These data were part of a full validation process, which allowed us to obtain the ISO1589 accreditation, granted by the Spanish National Accreditation and Control Entity (ENAC), for the panel.

Reviewer #4 (Remarks to the Author):

This study reports the results of a randomised phase II study that aimed to assess the efficacy and safety of continuous versus intermittent schedules of administration of Vemurafenib in combination with Cobimetinib, in previously untreated BRAFV600-mutation positive patients with unresectable locally advanced or metastatic melanoma. Beyond the initial trial outcomes in the study design, authors have conducted an exploratory translational sub-study in cell-free DNA using a subset of the patients. Authors should be commended for this well designed and well conducted study and the

concise and well written report. However I have few comments that I hope, will help to better interpret the results. In their study protocol (p 58), it is stated “With this sample size we would have a power of 80% to detect a 23% difference in the percentage of patients free of progression at 1 year...”. However, when looking at Figure 1A, we can see that the 12 month survival rates difference is much larger than 23% (about 60%-25% =35%), in addition the observed median between the two arms (16.2 versus 6.9) resulted in a much higher median difference compared to the expected median difference of 4 months (10-6 months) in the sample size calculation. Despite these findings that I found rather interesting, the main conclusion of the study is based on the single Log-rank test p-value. A known weakness of Log-rank test is it fails (no power) if the 2 hazard cross. Even though the hazard curves are not plotted I suspect the proportional hazard assumption does not hold for the PFS survival curves in Figure 1A (for the extended eFigure 3, the OS curves are crossing consequently the hazard crossed and the proportional hazard is violated). I would suggest to formally assess the proportional hazard assumption and consequently use more powerful test when the proportional hazard does not hold.

ANSWER: The p value obtained in the PFS analysis (p=0.07) was significant according to the design of the study that used a high α error level for sample size calculation (error $\alpha = 0.1$). Although with this design we had a 10% probability of incorrectly rejecting the true null hypothesis, the advantage was that the number of patients included and randomized was reduced compared with other conventional study designs. In our opinion, we should not focus on the P value alone to decide whether the experimental treatment arm is clinically different, because it is essential to consider the magnitude of treatment differences and the power of the study. In this study the magnitude of the difference between both schedules is clinically relevant and it is concordant with the previous data that showed detrimental results with the experimental schedule.

We have included information about the sample size calculation in the Supplementary Online Content in the Method Section:

*“Using the method described by Brookmeyer R., for an error $\alpha = 0.1$ and an error $\beta = 0.20$, it will be necessary to include **34** evaluable patients per treatment group, using Log-rank (Mantel-Cox) 2-sided test. With this sample size we would have a power of 80% to detect a difference of **23%** in the percentage of patients free of progression to 1 year (with an error $\alpha = 0.1$)”*

We have tried to modify the strength of our main conclusions, including a final sentence:

“These results come to confirm the detrimental effect of intermittent dosing, contrary to previous results in animal models arguing for the benefit of a drug holiday over the MAPK pathway inhibition.”

Minor comment: please include 1-year and 2-year survival rates in Figure 1a extended eFigure 3 as planned in the SAP.

ANSWER: 1-year and 2-year survival rates have been included in in Figure 1a extended eFigure 3.

The manuscript has been edited to include the requested responses according to the requirements. Two documents are submitted, one with revisions marked using “tracked changes” and a clean version with all changes accepted. Source data are provided with this paper.

We hope the manuscript will now be suitable for publication. On behalf of all coauthors, we thank you in advance for your time in considering our work for the Nature Communications.

Sincerely,

Maria Gonzalez Cao, M.D, PhD.

*Dr. Rosell Oncology Institute (IOR)
Dexeus University Hospital
C/ Sabino Arana, 5, 080028 Barcelona (Spain)
Tel: +34 93 546 0135
Email: mgonzalezcao@oncorosell.com*

Alfonso Berrocal, MD, PhD

*Oncology Department
Hospital General de Valencia
Av. De las tres creus, 46014 Valencia (Spain)
Tel. +34 963 13 18 00
Email: berrocal.alf@gmail.com*

Reviewers' Comments:

Reviewer #2:

Remarks to the Author:

The major criticisms have been adequately addressed and the manuscript is suitable for publication. Please review the final draft for typos.

Reviewer #3:

Remarks to the Author:

In general, all questions and suggestions were well answered and justified. The manuscript is suitable for publication.

Particularly, about the question and answer:

2. In the Extended data eTable 1. - Patient Characteristics there is no p value to analyze the homogeneous distribution of the two groups, please provide.

ANSWER: According to the reviewer recommendation, we have calculated the p value for each patient characteristic described in eTable1. We had not included in the eTable1 this p value following the CONSORT guidelines that, as far as we know, recommend that significance testing of baseline differences in randomized controlled trials should not be performed. Please, let us know if this information should be included in the final version of the supplementary appendix.

According the justification above, there is no necessity to change the table.

Reviewer #4:

Remarks to the Author:

Thanks for addressing all my queries

Response to Reviewers specific comments:

REVIEWERS' COMMENTS

Reviewer #2 (Remarks to the Author):

The major criticisms have been adequately addressed and the manuscript is suitable for publication. Please review the final draft for typos.

Response: The final draft of the manuscript has been reviewed for typos and language

Reviewer #3 (Remarks to the Author):

In general, all questions and suggestions were well answered and justified. The manuscript is suitable for publication.

Particularly, about the question and answer:

2. In the Extended data eTable 1. - Patient Characteristics there is no p value to analyze the homogeneous distribution of the two groups, please provide.

ANSWER: According to the reviewer recommendation, we have calculated the p value for each patient characteristic described in eTable1. We had not included in the eTable1 this p value following the CONSORT guidelines that, as far as we know, recommend that significance testing of baseline differences in randomized controlled trials should not be performed. Please, let us know if this information should be included in the final version of the supplementary appendix.

According the justification above, there is no necessity to change the table.

Response: Thank you

Reviewer #4 (Remarks to the Author):

Thanks for addressing all my queries

Response: Thank you

The manuscript has been edited to include the requested responses according to the requirements.

A final version of the manuscript as a Word document is submitted in two versions: one with revisions marked using "tracked changes" and a "clean version" with all changes accepted.

Source data are provided with this paper.

We have additionally attached:

- A revised author checklist describing our response to editorial requests
- Figure 1 divided into panels has been labelled with a lower-case, boldface 'a' and 'b' in the top left-hand corner. Panel Figure 2 has been provided.
- An updated checklist that verifies compliance with the research ethics and data reporting standards of the journal in PDF format.
- The final version of the Supplementary Information in one PDF file.

On behalf of all coauthors, we thank you in advance for accepting our work for publication in Nature Communications.

Sincerely,

Maria Gonzalez Cao, M.D, PhD.

Alfonso Berrocal, MD, PhD

*Dr. Rosell Oncology Institute (IOR)
Dexeus University Hospital
C/ Sabino Arana, 5, 080028 Barcelona (Spain)
Tel: +34 93 546 0135
Email: mgonzalezcao@oncorosell.com*

*Oncology Department
Hospital General de Valencia
Av. De las tres creus, 46014 Valencia (Spain)
Tel. +34 963 13 18 00
Email: berrocal.alf@gmail.com*